# The interplay between age structure and cultural transmission

**Anne Kandler** [1]*, **Laurel Fogarty**[1], **Folgert Karsdorp**[2]

**1** TICE lab, Department of Human Behavior, Ecology, and Culture, Max Planck Institute for Evolutionary Anthropology, Leipzig, Germany, **2** Meertens Institute, The Royal Netherlands Academy of Arts and Sciences, Amsterdam, The Netherlands

* anne_kandler@eva.mpg.de

**Data Availability Statement:** Code used to implement this approach is deposited at https://zenodo.org/badge/latestdoi/344132645.

**Funding:** The authors received no specific funding for this work.

## Abstract

Empirical work has shown that human cultural transmission can be heavily influenced by population age structure. We aim to explore the role of such age structure in shaping the cultural composition of a population when cultural transmission occurs in an unbiased way. In particular, we are interested in understanding the effect induced by the interplay between age structure and the cultural transmission process by allowing cultural transmission from individuals within a limited age range only. To this end we develop an age-structured cultural transmission model and find that age-structured and non age-structured populations evolving through unbiased transmission possess very similar cultural compositions (at a single point in time) at the population and sample level if the copy pool consists of a sufficiently large fraction of the population. If, however, an age constraint—a structural constraint restricting the pool of potential role models to individuals of a limited age range— exists, the cultural compositions of age-structured and non age-structured population show stark differences. This may have drastic consequences for our ability to correctly analyse cultural data sets. Rejections of tests of neutrality, blind to age structure and, importantly, the interaction between age structure and cultural transmission, are only indicative of biased transmission if it is known *a priori* that there are no or only weak age constraints acting on the pool of role models. As this knowledge is rarely available for specific empirical case studies we develop a generative inference approach based on our age-structured cultural transmission model and machine learning techniques. We show that in some circumstances it is possible to simultaneously infer the characteristics of the age structure, the nature of the transmission process, and the interplay between them from observed samples of cultural variants. Our results also point to hard limits on inference from population-level data at a single point in time, regardless of the approach used.

## Author summary

Age structure is undoubtedly a feature of human populations and empirical cultural evolutionary research has shown that the specifics of age structure can heavily influence how humans learn and transmit different cultural traits. Nevertheless, such demographic

**Competing interests:** The authors have declared that no competing interests exist.

properties are rarely included in theoretical models of human cultural evolution. This may present a problem, especially for model-based inferential analyses. In this paper, I show when the population-level signatures of age-structured and non age-structured populations—evolving through the process of unbiased transmission—show detectable differences and therefore when demographic properties must be included in models of cultural evolution. I demonstrate that a failure to do so may lead to erroneous conclusions about evolutionary processes, in particular processes of cultural transmission, underlying observed patterns of cultural change. To mitigate this risk I develop a generative inference framework combining cultural evolutionary and machine learning techniques. This framework also provides valuable information about hard limits on inference from sparse population-level data.

## Introduction

The populations of many species exhibit age structure in the sense that they consist, at any one time, of individuals of various ages. As such, there is a large body of theoretical literature addressing evolution in age-structured populations focusing on various topics from age-specific birth rates [1, 2], discrete age structures [3] and the evolutionary process [1, 4], life history evolution [5, 6], and mate choice [7] among many others (see [8], for a recent discussion of this literature).

In contrast, the much younger field of cultural evolution understands less about the ways in which age structure affects *cultural* evolutionary dynamics. Some theoretical work has shown that age structure mediates the kinds of cultural traits that can spread in populations characterised by different age structures and life histories [9], the ways in which transmission processes determine cultural diversity [10], and the evolution of cultural transmission mechanisms [11]. Work stemming from models developed by Carotenuto et. al. [12] has usefully applied age-structured demographic models to explain the maintenance and spread of cultural traits affecting human health and well-being [13].

Further, empirical work has shown that the way in which humans learn and transmit different kinds of cultural traits appears to be heavily influenced by age structure. For example, Hewlett and Cavalli-Sforza [14] showed that in the Central African Republic population of the Aka, over 80% of cultural traits they examined were learned vertically from parents. For other sets of cultural traits, learning may be from older unrelated individuals or from same-age peers [15]. Indeed, the definitions of many cultural transmission processes explicitly reference age. Much current theory, however, does not reflect the complexity of the relationship between transmission mechanisms and age structure. Many if not most models of cultural evolution assume that transmission occurs in well-mixed populations with no demographic or social structure. However, taken together, the mounting theoretical and empirical evidence suggests that the inclusion of age structure in modelling frameworks may be particularly important in the context of human cultural evolution.

In this paper we develop a simulation model to investigate the potentially important effect of age-structure on cultural evolution. We compare the cultural dynamic in age-structured and non age-structured populations which are evolving through unbiased cultural transmission. In particular, we are interested in understanding the effect induced by the interplay between age structure and the cultural transmission process—by allowing cultural transmission from individuals within a limited age range only. In the cultural evolutionary literature, 'unbiased transmission' has been defined as a process where "naive individuals randomly

choose a model" from whom to learn [16]. In other words, we use the term 'unbiased transmission' (also called 'linear transmission' by Boyd and Richerson [16]) to describe a situation where the probability of choosing any cultural variant to adopt is dictated only by its relative frequency in the pool of potential role models [16], p.134. This focus on unbiased transmission allows direct comparison with an extensive and well-developed body of theory addressing the evolution of selectively neutral cultural and genetic traits for non age-structured populations and facilitates quantification of the cultural differences between age-structured and non age-structured populations. We note that the age-structured models of cultural transmission developed in this paper do not seek to replicate a specific system. Rather, they are used to understand theoretically whether age structure and in particular, interactions between age structure and an unbiased transmission process can affect the dynamic of cultural change.

Mimicking the sparse data situation in many archaeological and anthropological case studies, we use the simulation model to explore the differences in the cultural composition, for example the numbers and frequencies of different variants of a cultural trait, of age-structured and non age-structured populations evolving through unbiased cultural transmission, at a single point in time. To this end we compare population- and sample-level properties of both types of populations. This comparison of theoretical expectations provides insights into first, the direct effect of age structure on the cultural composition and second, the compound effect of the age structure through its interaction with cultural transmission processes.

A second focus of the paper is concerned with the question of whether and how we can infer features of the cultural transmission process from population-level frequency data taken at a single point in time. Due to the sparse nature of the data, researchers often focused on the distinction between unbiased and biased transmission by using 'tests of neutrality'—in particular the Ewens-Watterson test (e.g. in [17–19], see below for more details). Following from the above definition of unbiased transmission, 'biased transmission' is assumed to describe situations where individuals use direct characteristics of cultural variants in order to choose the most adaptive variant to copy, use secondary characteristics (for example success or prestige) to choose a role model, or choose a variant disproportionately to its frequency (for example favouring common variants) [16], p.135. Put differently, 'biased transmission' describes any deviation from the assumption that individuals choose a cultural variant through random copying; they may have a preference for adopting one variant over another. We note that our conception of age-restricted pools of potential role models falls outside of this definition of biased cultural transmission. Individuals themselves choose through a process of random copying but from a restricted pool of role models. Nevertheless, restricted copy pools pose a structural constraint on copying which—to avoid confusion—we call an age constraint. Such a structural constraint, which could be, for example demographic, geographic, or normative in origin, may have the potential to influence the dynamic of cultural change.

Where a data set fails a test of neutrality, it is often concluded that the trait or variant in question is under some form of 'selection', or in other words, that the underlying transmission process deviates from random copying. However, such conclusions are problematic for a number of reasons (e.g. [20]). Notably, neutrality tests typically assume that populations have no significant structure or age structure and therefore cannot take into account potential interactions between age structure and the unbiased transmission process. Consequently, such a test might fail despite the fact that the process of cultural transmission remains unbiased (in the sense that naive individuals still choose role models at random) but, instead, because of the age constraint leading to a restriction on the copy pool. Thus, in order to confidently use such tests on human cultural data and correctly interpret their outputs, it is necessary to quantify the impact of violating the test's demographic assumptions—which in turn could induce structural

constraints. It may prove necessary, too, to develop new inferential procedures capable of delineating the effects of structural constraints from effects of biased transmission.

In this context, strides have been made in recent years to adapt an inferential approach developed in population genetics which goes beyond the hypothesis testing framework (e.g. [21–25]). The generative inference framework simultaneously evaluates the consistency of a number of cultural transmission processes with the available data while also accounting for demographic and cultural properties of the system considered. In general, the approach consists of two main steps. The first step comprises the development of a generative model, capturing the main cultural and demographic dynamics of the cultural system considered, to produce pseudo-data, in our case population-level frequencies of different variants conditioned on an assumed cultural transmission process. The second step uses Bayesian techniques to statistically compare theoretical predictions and empirical observations and to derive conclusions about which (mixtures of) transmission processes are consistent with the observable frequency data (and which are not). Naturally, the accuracy of the inference results depends on the appropriateness of the generative model, i.e. how well the model reflects the dynamic of cultural change of the system considered. Consequently, the results of our analyses will indicate whether and when demographic properties of the population such as age structure need to be included in generative models.

We develop a generative inference approach based on our age-structured cultural transmission model and machine learning techniques (see also [26] for an application to simulated archaeological data) and show that in some circumstances it is possible to simultaneously infer the characteristics of the age structure, the nature of the transmission process, and the interplay between them from observed samples of cultural variants. This approach also allows us to explore the hard limits to such inferential exercises, given the temporal resolution of a single point of the data at hand.

# 1 Methods

## 1.1 Age-structured cultural transmission model

We consider a population of size $N$. Each individual in the population is characterised by its age and the cultural variant it has adopted. At each time step, individuals age and may die with probability $p_{death}$. We assume that the population size remains constant over time which implies that the birth and death processes are not independent, and the number of new individuals entering the population in any time step is equal to the number of deaths. On average, this number is $Np_{death}$. Thus, the age structure of the population is determined by the parameter $p_{death}$: the smaller $p_{death}$ the fewer newborn individuals enter the population at age 0 and the longer the lifespan of an individual in time steps.

Each newborn individual is initially naive with respect to the cultural variant. They can adopt a variant from a role model chosen at random from a pool of potential models. This pool is determined by the parameter $c_{thresh}$ which describes the maximum age of the individuals it contains. In what follows, we contrast the two extreme scenarios: (i) the 'ALL' scenario with $c_{thresh} = a_{max}$, where $a_{max}$ describes the age of the oldest individual in the population which can change over time. In this case, the pool of potential models is the whole population of living individuals. And (ii) the '1' scenario with $c_{thresh} = 1$, where naive individuals choose role models from all individuals of age 1, or in other words copy only the most recently transmitted cultural variants.

The cultural variants have no effect on the survival or reproductive capacity of the individuals or the variant type itself; they are selectively neutral. Under unbiased transmission, as defined above, a cultural variant type $i$ is chosen to be copied by a naive individual with

probability

$$p_i = \frac{n_i^P}{N^P}(1-\mu) \qquad (1)$$

where $n_i^P$ describes the number of instances of type $i$ in the variant pool of size $N^P$. Any restriction of the copy pool induces a structural constraint in the form of an age constraint; the 'ALL' scenario represents a situation without any constraints whereas the '1' scenario a situation with strong constraints. Further, the transmission process is faithful with probability $(1-\mu)$. With probability $\mu$, a new, not previously seen cultural variant is introduced into the system. To allow for consistency with the classical Wright-Fisher infinite alleles model [27, 28] we assume that reproduction and death occur simultaneously, i.e. a naive individual can choose a role model from all individuals in the appropriate copy pool that were present in the previous time step.

To include cultural transmission processes other than unbiased transmission we must specify an appropriate probability for choosing variant type $i$. Here we focus on frequency-dependent transmission and alter Eq (1) to

$$p_i = \frac{(n_i^P/N^P)^{1+b}}{\sum\limits_{j=1}^{k}(n_j^P/N^P)^{1+b}}(1-\mu), \qquad (2)$$

where $k$ describes the number of variant types present in the population and $b$ the strength of the frequency-dependent bias. For $b < 0$ we model negative frequency-dependent bias and for $b > 0$ positive frequency-dependent bias. As above, any restriction of the copy pool induces an age constraint.

To obtain information about the cultural composition of the population, we let each simulation reach its steady state and then extract the frequencies of all variant types present in the population as well as in samples of different sizes. Throughout the manuscript we consider the following parameter ranges $N = 10^5$, $\mu = 5 \cdot 10^{-4}$, $p_{\text{death}} = 0.02, 0.03, 0.04, 0.05, 0.1$ and $c_{\text{thresh}} = 1, a_{\text{max}}$. All results shown are generated based on 100 independent simulations.

## 1.2 Test of neutrality

**1.2.1 Ewens-Watterson test.** The Ewens-Watterson test is a test for selective neutrality developed by population geneticists Ewens and Watterson [29–31] and generalised by Slatkin [32, 33]. The test evaluates whether an observed sample of size $n$ could have been drawn from a population whose composition has evolved through a Wright-Fisher dynamic. Ewens [29] showed that under neutrality for a given number of variant types $k$, the probability of obtaining a specific configuration of types, $(n_1, n_2, \ldots, n_k)$ with $\sum_{i=1}^{k} n_i = n$, in the sample is given by

$$P(n_1, n_2, \ldots, n_k | n, k) = \frac{n!}{|S_n^k| k! n_1 \cdot \ldots \cdot n_k}, \qquad (3)$$

where $|S_n^k|$ denotes the unsigned Stirling number of first kind [29]. Ewens [29] and Watterson [30, 31] used Eq (3) to propose a test of neutrality based on the observed diversity in the sample. In other words, this test evaluates whether the level of diversity observed in a sample of size $n$ could have been drawn from a population distributed according to (3).

Slatkin [32, 33] generalised this approach based on Fisher's exact test by calculating the tail probability of the observed configuration $\mathbf{n}_0 = (n_{0,1}, n_{0,2}, \ldots, n_{0k})$

$$P_E = \sum_{\{\mathbf{n}_j : P(\mathbf{n}_j|n,k) \leq P(\mathbf{n}_0)\}} P(\mathbf{n}_j|n, k).$$

The set $\{\mathbf{n}_j : P(\mathbf{n}_j|n, k) \leq P(\mathbf{n}_0)\}$ contains all possible configurations $\mathbf{n}_j = (n_{j,1}, n_{j,2}, \ldots, n_{j,k})$ with $\sum_{i=1}^{k} n_{j,i} = n$ and $k$ variant types as observed in $\mathbf{n}_0$ whose probabilities $P(\mathbf{n}_j|n, k)$—given by Eq (3)—are smaller than the probability $P(\mathbf{n}_0)$ of the observed data $\mathbf{n}_0$ being neutral. However, for large sample sizes it is usually impractical to determine all possible configurations. In this case, we can sample a large number of configurations from Eq (3) (see [34] and S1 Text, section 2 for the description of the sampling algorithm). We then count the configurations with probabilities equal or less than the probability of the observed configuration $\mathbf{n}_0$. This provides a $p$-value suitable for a two-tailed hypothesis test with significance level $\alpha$.

To summarise, by determining where the observed sample $\mathbf{n}_0$ is situated in the distribution of neutral configurations generated by Eq (3), one can potentially draw conclusions about the evolutionary forces, i.e. the presence or absence of selective forces, that shaped the composition of sample $\mathbf{n}_0$. However, the neutral expectation in Eq (3) was derived based on a Wright-Fisher dynamic and explicitly assumes a population without age structure which is in stark contrast to real world populations generating most cultural data sets. We are interested in understanding whether the Ewens-Watterson test can fail in situations where transmission is unbiased but the population is age-structured, or put differently, whether the Ewens-Watterson test can detect a structural constraint in form of an age constraint in the data. If this is the case, researchers need to be careful in interpreting the test result: rejections are not *per se* indicative of biased transmission, i.e. a deviation from the random copying hypothesis but can—as in our case—point to constraints induced by demographic properties unaccounted for in the Wright-Fisher model.

**1.2.2 Machine learning approach.**   In this section we introduce a machine learning (ML) approach that reformulates the neutrality test described above as a binary classification problem. This approach is based on research on supervised classification methods [35], and we apply it here to train a system that can automatically learn how to distinguish samples taken from populations evolving through unbiased transmission and those taken from populations evolving through biased transmission based on simulated training material. This method has recently been applied to study aspects of language change [36].

At the heart of this approach is the generation of appropriate training data which allow the system to learn which properties of the data can be associated with which label, in our case with the labels 'unbiased' and 'biased'. Or in ML terminology: supervised classification systems are trained based on pairs of features $x_i$ and corresponding labels $y_i$. Given a data set $D$ with $n$ of these pairs, $D = \{(x_1, y_1), (x_2, y_2), \ldots, (x_n, y_n)\}$, the goal is to learn a mapping function $f$ to associate the features $x_i$ with the labels $y_i$, i.e., $y_i = f(x_i)$. In what follows, we will first describe our chosen feature representation, and successively provide details about the supervised classification system.

We describe the features of our data, i.e. random samples of the cultural composition of the population, by a family of diversity measures called Hill numbers (with $q \neq 1$) [37–40]

$$^q D((p_1, \ldots, p_k)) = \left( \sum_{i=1}^{k} p_i^q \right)^{(1/1-q)} \tag{4}$$

where $p_i$ represents the relative frequency of variant type $i$, and $q$ the sensitivity to the relative

frequencies of variant types. Using simple algebraic operations, Hill numbers can be transformed into well-known diversity indices. For example, $^0D$ is equal to the richness, $k$, of a sample, since no weight is given to the relative frequency of variant types. Put differently, with $q = 0$, maximum weight is given to rare types. With $q = 1$ all types are equally weighted by their relative frequency. Note however that $^1D$ is undefined, but the limit exists, which is equal to the exponential of Shannon entropy (cf. [39, 40]). With $q > 1$, common variant types are disproportionately weighted, giving less weight to rare types for increasing values of $q$. For instance, the Hill number of order 2 is equal the inverse of the Gini-Simpson index (or expected heterozygosity) which describes the probability that two variants randomly chosen from the sample are of the same type. One advantage of Hill numbers is that they are all expressed in units of *effective number of variants*: the number of equally abundant variant types that would be needed to lead to the same value of the diversity measure. Diversity is thus measured on the same scale, which allows us to chart "diversity profiles", representing the diversity at different orders $q$. As our feature representation, then, we compute diversity profiles with $q$ in the interval [0, 3] and a step size of 0.25 (see [39, 40]).

To be able to automatically associate features of data with the two labels 'unbiased' and 'biased', we use random forest classifiers [41] as implemented in the Scikit-learn toolkit [42]. Random forest classifiers belong to the family of ensemble classifiers, that make predictions by combining or "bagging" the output of $T$ sub-classifiers. More specifically, random forests combine a multitude of $T$ decision tree classifiers (thus forming a "forest") to construct a meta classifier $\hat{f}$. These decision trees are supervised learning methods that aim to infer simple "if-then-else" decision rules from the features. These rules can either lead to subordinate decision rules or to leaf nodes representing the class labels 'biased' or 'unbiased' (e.g., "if $^1D \leq 1 \rightarrow y_i = 1$", and see the example trees in Fig 1C). Random forests are "random" in two key respects. First, each decision tree in the ensemble is built on the basis of a random sample (with replacement) of the training data, $X'$, and corresponding labels $Y'$. Second, when constructing trees, splits are formed on the basis of a random selection of input features (i.e. a random selection of values from the diversity profile). These two perturbation strategies aim to prevent the tendency of individual decision trees to overfit the training data and help decorrelating the trees. In the binary case, then, random forest classifiers predict the label $y_i$ of a particular sample $x_i$ by taking the average of the binary predictions of the $T$ decision trees ($\hat{f}(x_i) = \frac{1}{T}\sum_{t=1}^{T} f_t(x_i)$) and choosing the majority class as the predicted label.

To train the random forest classifier (i.e. to approximate the mapping function $f$) we need sufficient training data. For that we use the age-structured cultural transmission model described in section 1.1 and generate 10,000 diversity profiles under unbiased and biased (i.e. frequency-dependent) transmission as well as under different demographic and cultural scenarios (see Figs J,K in S1 Text for a justification of this number of training samples). In more detail, we (i) sample parameter values for each simulation from the following distributions

$$b \sim \mathcal{N}(0, 10^{-3}), \ \mu \ \sim \mathcal{U}(10^{-5}, 10^{-3}), \ p_{\text{death}} \ \sim \mathcal{U}(0.02, 0.1) \ \text{and} \ c_{\text{max}} \sim [1, a_{\text{max}}], \quad (5)$$

(ii) produce the frequency distribution of cultural variant types at steady state, (iii) take random samples of a particular size, and (iv) calculate the diversity profiles according to (4). We train separate classifiers for different sample sizes $n$ with $n \in \{100, 500, 1000, 2000\}$. Importantly, for each simulated diversity profile we know its label, i.e. we know whether it has been generated in an unbiased way with $b = 0$ or in a biased way with $b \neq 0$. This means we can label all samples taken from populations—with any age constraints—where individuals choose their role models through random copying as 'unbiased'. For the complete pipeline of the algorithm, from data generation to classification, see Fig 1. To evaluate how well this approach is

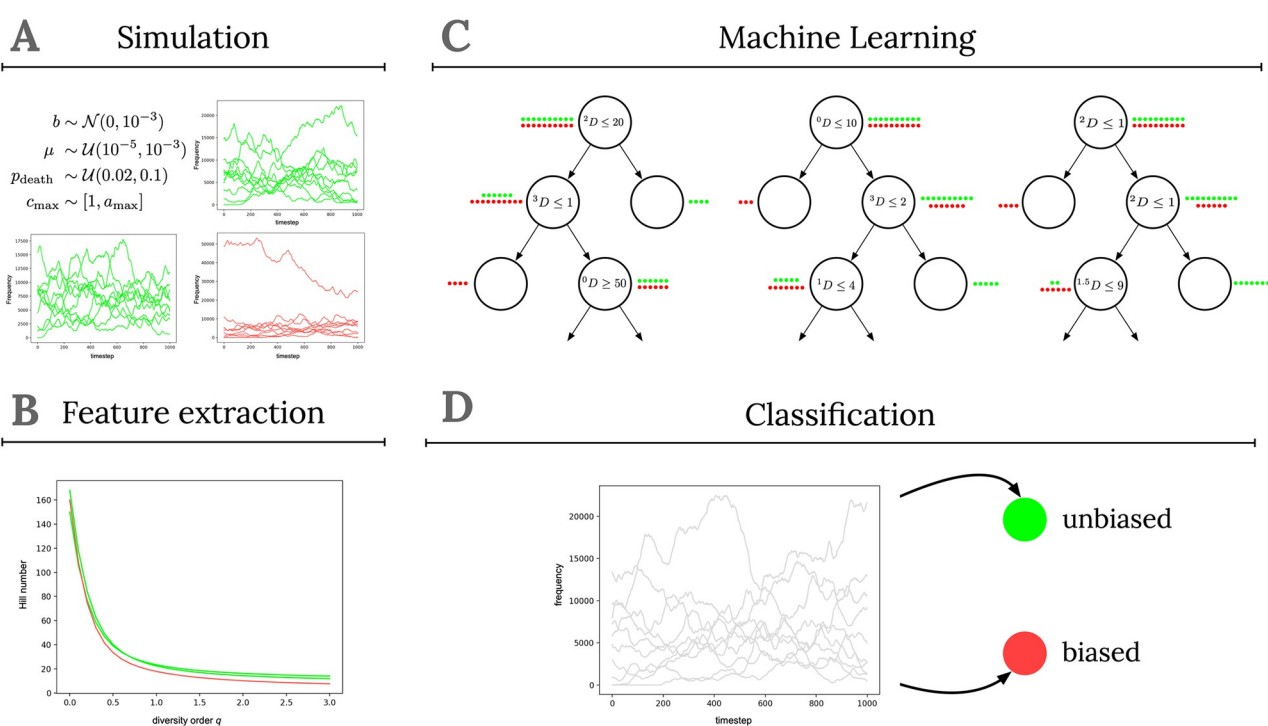

**Fig 1. The four components of the Machine Learning approach.** Panel A depicts the data generation step, in which we employ the simulation model to generate a large number of training samples. On the basis of the generated samples, features are extracted in the second component (see panel B). The features and corresponding labels are then used to train a supervised Machine Learning system (panel C). Finally, in D the trained system is used to classify new, unseen samples as either 'unbiased' or 'biased'.

able to detect signatures of unbiased transmission under various age structure assumptions, we use in the next section the developed algorithm and classify diversity profiles generated under fixed parameter constellations and calculate for each constellation the fraction of samples classified as biased.

**1.2.3 Comparison between Ewens-Watterson test and the ML approach.** An important conceptual difference between the Ewens-Watterson test and the binary classification ML setup is how 'bias' is formulated and defined. The Ewens-Watterson test, in the form used here, does not test against a particular alternative hypothesis and instead uses the probability of each sample configuration to indicate which regions of the sample space allow the rejection of the hypothesis that the sample was taken from a population whose composition has evolved through a Wright-Fisher dynamic [32]. By contrast, the ML classification approach aims to distinguish between two hypotheses—unbiased and biased—but what is considered 'biased' is defined by all biased processes that are used to generate the sample pool.

We compare the results of both approaches described above when applied to the same data sets generated through a Wright-Fisher dynamic (i.e. without age structure or equivalently $p_{\text{death}} = 1$ in the simulation framework developed above), but with different strengths of frequency-dependent bias $b$ (see copy probability given by Eq (2)). In other words, we ask how well both tests can distinguish between unbiased and frequency-dependent cultural transmission initially in populations without age structure. To do this we simulate 100 populations for each value of the parameter $b$, then extract samples of different sizes after the steady state has been reached, apply the neutrality tests and record how many times the hypothesis of unbiased

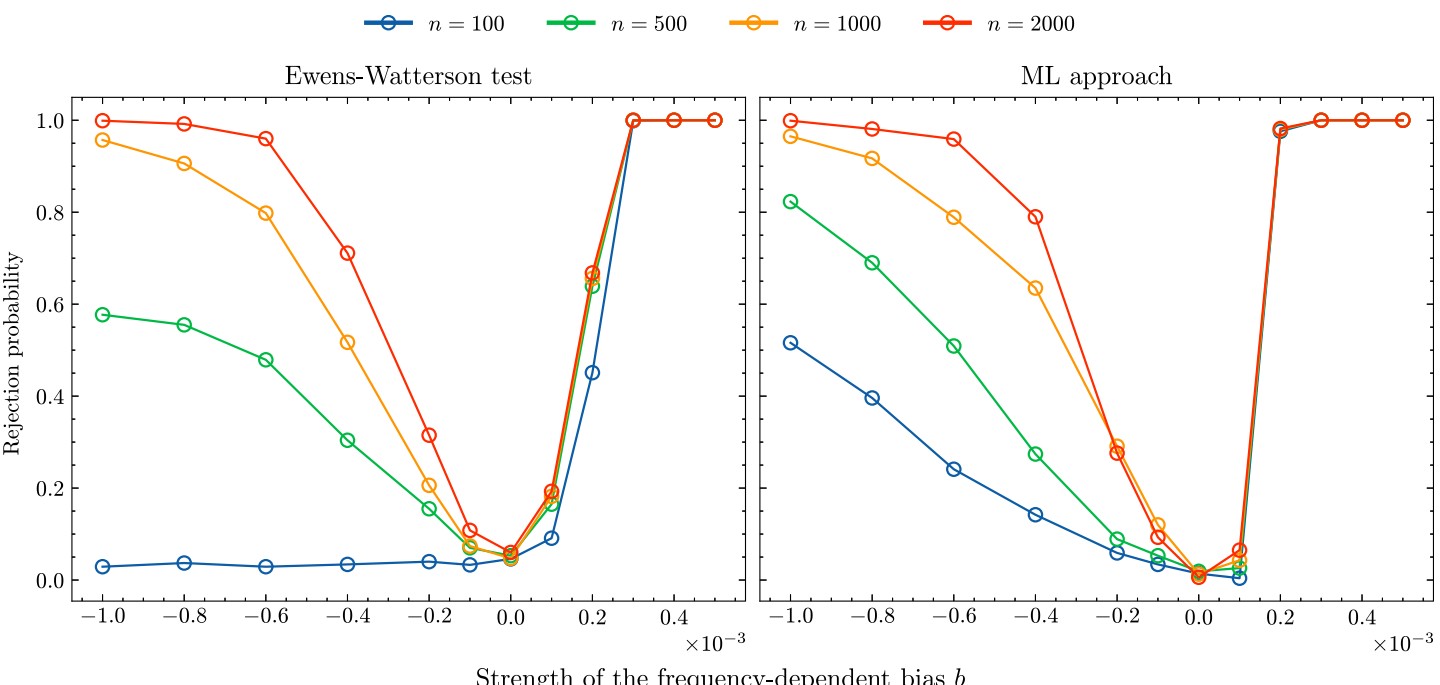

**Fig 2. Rejection probability of a sample taken from an WF population with $N = 10^5$, $\mu = 5 \cdot 10^{-4}$.** Left: Ewens-Watterson test and right: ML approach. The different coloured lines represent different sample sizes: 100 (blue lines), 500 (green lines), 1000 (orange lines), 2000 (red lines).

transmission has been rejected. For the ML-approach, this amounts to counting the number of times samples are classified as biased.

The left panel in Fig 2 shows these rejection probabilities for the Ewens-Watterson test. For most sample sizes we see a U-shaped rejection profile where unbiased transmission is rejected for $b = 0$ with a probability of roughly 0.05, the chosen significance level. With the exception of sample size $n = 100$, the rejection probabilities increase the further the value of $b$ deviates from 0. While a weak frequency-dependent bias does not generate sample compositions substantially different from unbiased transmission, and, consequently, is detected only with a small probability, a strong frequency-dependent bias is reliably detected if the sample size is sufficiently large.

Interestingly, we see a difference in the rejection probabilities for negative and positive frequency-dependent cultural transmission due to the fact that positive $b$-values affect the frequency composition of the population more strongly than negative $b$-values of the same magnitude (see Fig F in S1 Text showing the variant abundance distributions for $b = 0$; 0.005; $-0.001$). The reason for this difference are the different targets of the biases: a positive frequency-dependent bias selects for high frequency variants (and consequently selects against low-frequent variants) while the opposite is true for a negative frequency-dependent bias. Consequently, for a negative frequency-dependent bias to become "visible" it has to overcome the effect of drift and keep low-frequency variants in the population at a higher rate than expected under unbiased cultural transmission.

The inability to detect negative frequency-dependent bias for sample size $n = 100$ (blue line in Fig 2, left panel) is rooted in a sampling problem: $n$ is too small to adequately reflect the properties of the population. We demonstrate this by comparing the shapes of Ewens sampling distribution (3) for a fixed number of variant types, $k$, obtained by (i) sampling from Eq (3)

and (ii) sampling from populations undergoing Wright-Fisher dynamics with the negative frequency dependent bias of $b = -0.001$ ($b = -0.001$ is the strongest bias considered and we would expect the differences between unbiased and biased transmission to be largest). As the normalisation terms $n!/(|S_n^k|k!)$ are identical in both scenarios we focus on the comparison of distributions of $1/\prod_{i=1}^{k} n_i$. For $n = 100$ both distributions are very similar (see Fig Ga in S1 Text). In particular, the support of the distribution generated through simulating Wright-Fisher dynamics with $b = -0.001$ is a subset of the support of the distribution generated through sampling from Eq (3). This is caused by the high number of variant types in the samples generated by a negative-frequency dependent bias ($k = 83$ variant types were generated most frequently for $n = 100$ in scenario (ii)). To replicate this number by unbiased transmission through Eq (3), a relatively high innovation rate has to be assumed implicitly leading to very similar sample-level statistics for samples with $k = 83$ and $n = 100$ for scenarios (i) and (ii) (see Figs Ha-c in S1 Text). However, if the sample size increases, a large proportion of samples in scenario (ii) generate values of $1/\prod_{i=1}^{k} n_i$ that are too small as to be generated by scenario (i) (see Fig Gb in S1 Text). Now the frequency-dependent bias generates samples with $k = 240$ variant types most frequently which leads to detectable differences in sample-level statistics (see Figs Hd-f in S1 Text)—samples generated by negative frequency-dependent transmission are more evenly distributed.

The right panel in Fig 2 shows the rejection probabilities (for the same data sets) of the ML approach. The ML approach produces results similar to the Ewens-Watterson test, with the difference that a (strong) negative frequency-dependent bias can be detected also for small sample sizes. The reason for this difference is rooted in the fact that the ML approach is not based on Ewens sampling distribution but the explicit comparison between samples generated by biased—frequency-dependent—transmission and unbiased transmission. Therefore biased and unbiased samples do not have to contain the same number of variant types and differences in these numbers as well as in other sample-level statistics as measured by the Hill numbers $^qD$ provide valuable information for the inference analysis even for small sample sizes. Additionally, the ML approach provides the flexibility to assign different weights to different $^qD$ numbers as appropriate.

To probe which $^qD$ values are most predictive, we compute the 'feature importances' of the trained classifiers using a permutation approach. Permutation-based importance is defined as the decrease in prediction performance—here measured in terms of accuracy—on a held-out development data set when the values of a single feature are randomly shuffled [41]. Thus, features that result in a larger performance drop are taken to be more important. Fig 3 shows the feature importance given to $^qD$, $q = 0, 0.25, 0.5, \ldots, 3$ for different sample sizes and it is obvious that the importance profile for $n = 100$ differs from the ones for $n = 500, 1000, 2000$.

Lastly, we note that the ML approach does not assume a pre-defined significance level like the Ewens-Watterson test. The probability with which an unbiased sample is rejected as inconsistent with the hypothesis of unbiased transmission is a reflection of the level of equifinality in the data. In other words, it tells us how likely an unbiased transmission process is generating a sample with a diversity profile that is indicative of a biased transmission process with $b \neq 0$.

## 2 Results

### 2.1 Influence of age structure on neutral dynamics

**2.1.1 'ALL' scenario.** The 'ALL' scenario allows naive individuals to choose their role model from the entire population of informed individuals. To understand the impact of age

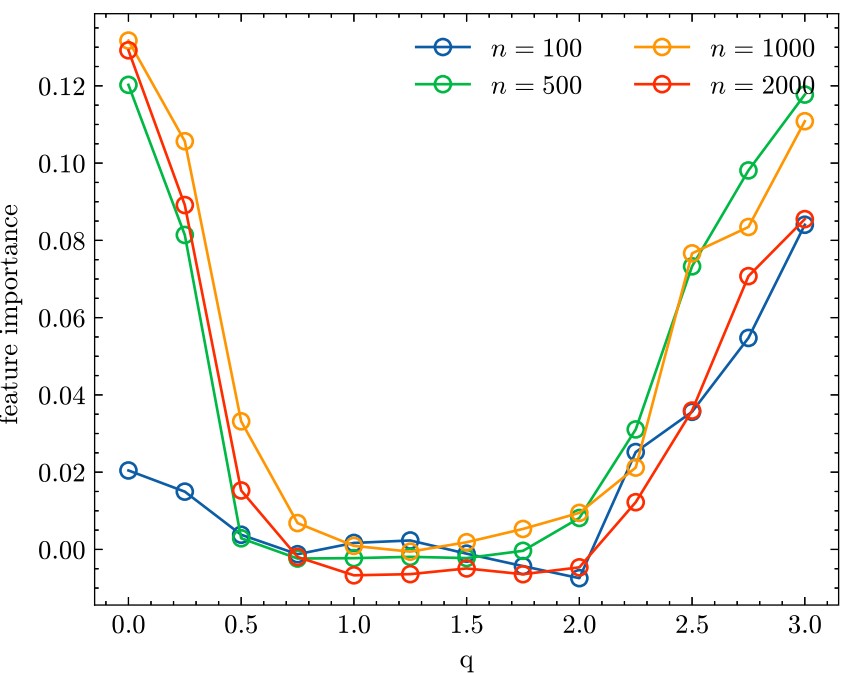

**Fig 3. Permutation importance of $^qD$ values for inference based on samples of size $n = 100$ (blue line), $n = 500$ (orange line), $n = 1000$ (green line), $n = 2000$ (red line).**

structure on the cultural composition in this scenario we first calculate the effective population sizes of age-structured neutral populations for various $p_{\text{death}}$-values (shown in Table 1 and see S1 Text, section 4 for details of how to calculate effective population sizes). Those effective population sizes are, as expected, larger than the effective size of the Moran model (which we obtain for $p_{\text{death}} \rightarrow 1/N$), i.e. larger than $\sim N/2$, and increase with increasing $p_{\text{death}}$. Given the size and relatively small differences between the effective population sizes for different $p_{\text{death}}$-values we do not expect large variations in the cultural composition of the age-structured populations; the analyses below confirm this intuition.

In S1 Text, section 1.3 we show that cultural composition at the population and sample level do not differ greatly for different values of $p_{\text{death}}$ and mostly coincide with their Wright-Fisher approximations with population sizes $N = N_e$ given in Table 1. While we observe very similar distributions for age-structured populations and their Wright-Fisher approximations for the level of cultural diversity, as measured by the heterogeneity index, and for the frequency of the most common variant type in the population (as expected from our definition of effective population size) we do see differences in the variant abundance distributions and the numbers of variant types present in the population. In particular, the variant abundance distributions of the age-structured populations considered here, i.e. with $p_{\text{death}}$-values between 0.02 and 0.1,

**Table 1. Effective population size of the age-structured neutral models with various $p_{\text{death}}$-values and $N = 10^5$.** Details on the calculation of those effective sizes can be found in S1 Text, section 4.

|  | $p_{\text{death}} = 0.02$ | $p_{\text{death}} = 0.03$ | $p_{\text{death}} = 0.04$ | $p_{\text{death}} = 0.05$ | $p_{\text{death}} = 0.1$ |
|---|---|---|---|---|---|
| 'ALL' | 50392 | 50757 | 51020 | 51282 | 52632 |
| '1' | 2000 | 3000 | 4000 | 5000 | 10000 |

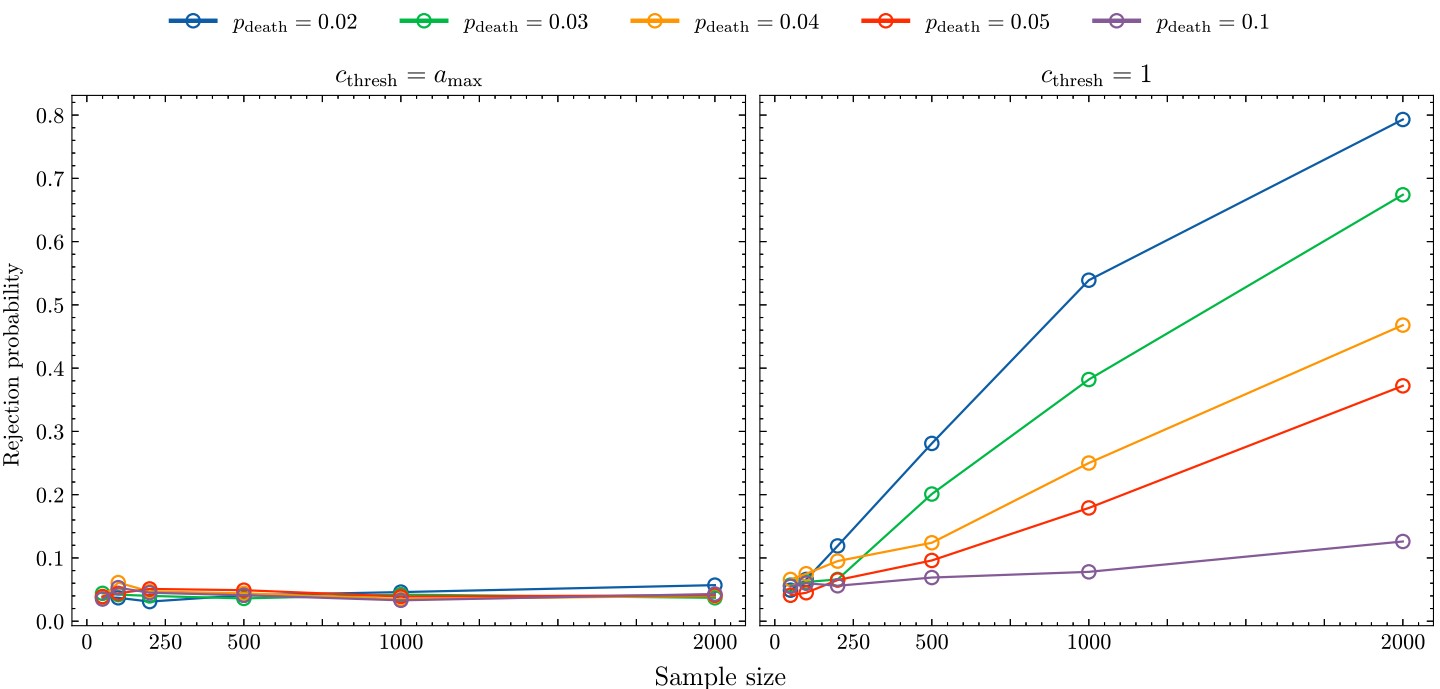

**Fig 4. Rejection probability of the Ewens-Watterson test for samples taken from an age-structured population with $N = 10^5$, $\mu = 5 \cdot 10^{-4}$, $b = 0$.** Left: $c_{thresh} = a_{max}$; right: $c_{thresh} = 1$. The different coloured lines represent different values of $p_{death}$: $p_{death} = 0.02$ (blue lines), $p_{death} = 0.03$ (green lines), $p_{death} = 0.04$ (orange lines), $p_{death} = 0.05$ (red lines), $p_{death} = 0.1$ (purple lines).

conform to the expectations of the Moran model and not to the expectations of the Wright-Fisher model (see Fig Ae in S1 Text). So the presence of an age structure leads to deviations from the Wright-Fisher model at the population level and those deviations resemble the ones expected between the classical Wright-Fisher and Moran model. However, we see that those deviations do not percolate to the sample level. The distributions of all sample-level statistics of age-structured and non-age-structured populations are very similar (see Fig B in S1 Text).

The left panel in Fig 4 shows the result of applying the Ewens-Watterson test described in Section 1.2 to 1000 samples of size $n = 50$; 100; 500; 1000; 2000 randomly drawn from age-structured populations with $c_{thresh} = a_{max}$ and different $p_{death}$-values. We recorded the fractions of populations classified as biased and the results show, again, only very few differences between age-structured populations and their Wright-Fisher approximations at the sample level. Around 5% of the age-structured populations, which equals the chosen significance level $\alpha$, are rejected and, consequently, considered inconsistent with Ewens sampling distribution (3). This implies that if the copy pool consists of the whole population there is little detectable difference in the sample-level dynamic of unbiased cultural transmission in age-structured and non-age-structured populations.

**2.1.2 '1' scenario.** The '1' scenario allows naive individuals to choose their role models from the last time step only, i.e. from all informed individuals of age 1. Table 1 illustrates that, compared to the 'ALL' scenario, the effective population sizes are smaller and the relative differences between $p_{death}$-values are higher. The number of naive individuals entering the population is, in both scenarios, on average $Np_{death}$ but while in the 'ALL' scenario the size of the copy pool stays constant for all $p_{death}$-values, it changes for the '1' scenario to, on average, $Np_{death}$.

In S1 Text, section 1.4 we show that, intuitively, the cultural composition at the population and sample level differ for different values of $p_{death}$: the lower the $p_{death}$-value, the lower is the number of variant types in the population and the higher the frequency of the most common variant type in the population. This subsequently leads to more culturally homogeneous populations. These differences arise in part because the number of innovations depends on $p_{death}$, on average $N p_{death} \mu$ innovations per time step, and a higher amount of drift due to the smaller size of the copy pool for lower $p_{death}$ values.

More generally, we observe a similar behaviour as in the 'ALL' scenario when comparing the distributions of the population-level statistics with their Wright-Fisher approximations with population sizes $N = N_e$ given in Table 1. While the level of cultural diversity and the frequency of the most common variant type show very similar distributions we see stark differences between age-structured and non age-structured populations in the variant abundance distributions and the numbers of variant types present, especially for small $p_{death}$-values. Importantly, these differences are sufficiently large to be detectable on the sample level. Mechanistically they are caused by the accumulation of variants in the non-reproductive population. In other words, individuals can only act as a role models at age 1 but they nevertheless survive, and consequently contribute to the cultural composition of the population, for a potentially long period of time. Fig D in S1 Text shows that the number of variant types contained in age group 0, i.e. the group of all individuals of age 0, after cultural transmission resembles the number of variant types contained in the Wright-Fisher approximations with $N = N_e$. Therefore the differences in the variant abundance distributions and in the number of types is determined by the types that are present in the population but not in age group 0.

The right panel in Fig 4 shows the results of applying the Ewens-Watterson test to 1000 samples of size $n = 50; 100; 500; 1000; 2000$ randomly drawn from age-structured populations with $c_{thresh} = 1$, i.e. with a strong age constraint, and different $p_{death}$-values. Age-structured populations are now considered inconsistent with Ewens sampling distribution (3) and therefore not classified as unbiased for a good fraction of the simulations, especially if $p_{death}$ is small and the sample size is high (We note that for smaller sample sizes we observe a sampling problem similar to the one described in section 1.2.3.).

The reason for this behaviour is partly rooted in the different number of variant types between the age-structured populations and their Wright-Fisher approximations. Age-structured populations with a strong age constraint generate a higher number of variant types and the samples generated by Ewens sampling distribution (3) for these numbers are characterised by more even distributions with less probability mass given to a single variant type. So, if the copy pool consists of informed individuals of age 1 only, there are detectable difference in the sample-level dynamic of unbiased cultural transmission in age-structured and non-age-structured populations.

To understand the dynamic for values of $c_{thresh}$ in between the two extremes considered above of 1 and $a_{max}$ we determined in Fig 5 the rejection probabilities of the Ewens-Watterson test for $c_{thresh} = 5; 10; 20$. The inclusion of individuals up to age 5 in the copy pool reduces rejection probabilities. Increasing the $c_{thresh}$ value further results in similar patterns as seen in the 'ALL' scenario described in section 2.1.1.

To summarise, the existence of an age structure *per se* may not influence the dynamic of unbiased cultural transmission greatly. In our model, it depends mainly on how the transmission process interacts with the age structure, i.e. whether the 'ALL' scenario models contains individuals of a limited age range only. Such an age constraint will generate differences in the cultural composition of age-structured and non age-structured populations. Consequently, if a sample is taken from an age-structured population, rejection of the hypothesis of unbiased transmission by the Ewens-Watterson test cannot be readily interpreted as

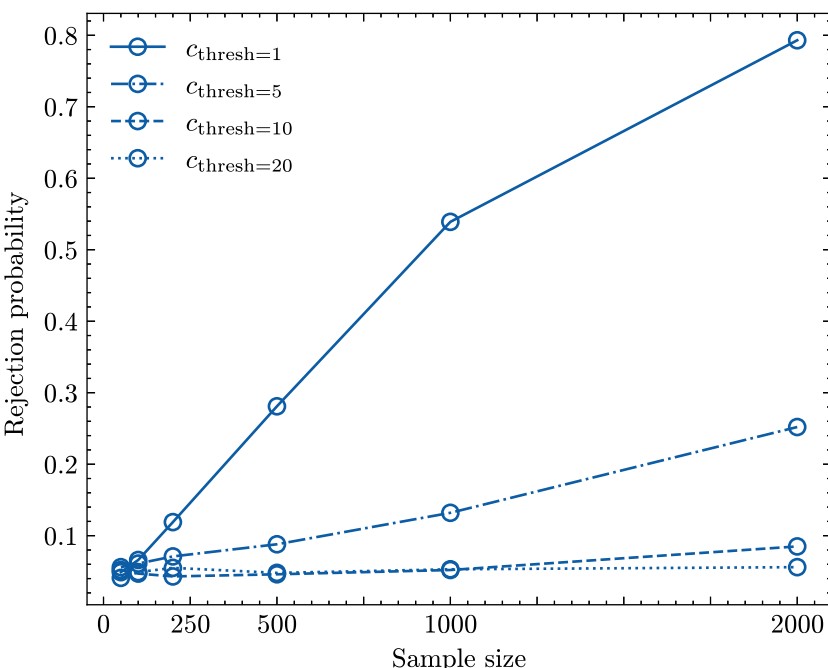

**Fig 5. Rejection probability for samples taken from an age-structured population with $N = 10^5$, $\mu = 5 \cdot 10^{-4}$, $p_{death} = 0.02$ and $c_{thresh} = 1$ (solid line), $c_{thresh} = 5$ (dash-dotted line), $c_{thresh} = 10$ (dashed line), $c_{thresh} = 20$ (dotted line).**

indicative of biased transmission—it may, instead, be indicative of the existence of an age (or other demographic) constraint on the copy pool. In other words, only in situations where it is known that the copy pool consists of a large fraction of the population can we infer biased transmission from the rejection of the Ewens-Watterson test (at least in the situation considered in this paper). As this knowledge is rarely available for specific empirical case studies, we explore in the next section whether the Machine Learning framework introduced in section 1.2.2 can simultaneously account for unknown demographic and cultural properties of the population and still accurately infer underlying processes of cultural transmission from data at a single point in time.

## 2.2 Accounting for a—Potentially unknown—Age structure

Here, we assume that an age structure exists but we do not have information about its depth nor its interaction with cultural transmission processes. Our aim is still to infer whether observed (sample-level) frequencies of cultural variants are consistent with the hypothesis of unbiased transmission, i.e. whether individuals choose a role model at random, and to do that we use the ML approach described in section 1.2.2.

We model the non-neutral, i.e. biased transmission condition by assuming frequency-dependent cultural transmission with different strengths $b$ (see Eq (2)). So, our training data provide information about the diversity profiles (4) under a large range of cultural transmission ($b$, $\mu$, $c_{thresh}$) and demographic ($p_{death}$) scenarios whereby the model parameters $b$, $p_{death}$, $\mu$ and $c_{thresh}$ cover the ranges described in (5).

We start by analysing the data sets used to generate Fig 5. Fig 6 shows the rejection probabilities obtained from the ML approach and we clearly see that, especially for small $c_{thresh}$-values, those probabilities are smaller compared to Fig 5. Further, the rejection probabilities do not exhibit a strong dependence on sample size as seen for the Ewens-Watterson test (cf. Fig

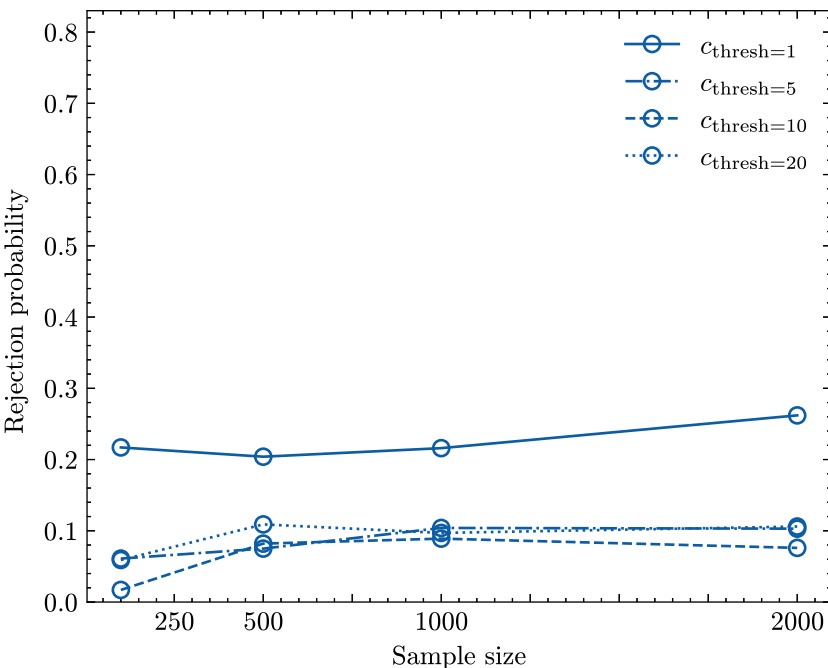

**Fig 6. Fraction of neutral samples classified as non-neutral (by the ML approach) taken from an age-structured population with $N = 10^5$, $\mu = 5 \cdot 10^{-4}$, $b = 0$, $p_{death} = 0.02$ and $c_{thresh} = 1$ (solid line), $c_{thresh} = 5$ (dash-dotted line), $c_{thresh} = 10$ (dashed line), $c_{thresh} = 20$ (dotted line).**

5). However, we also observe that the rejection probability is slightly increased for higher $c_{thresh}$-values: the rejection probabilities are around 0.1. This reflects the level of equifinality in the data contained in the training pool. In 10% of the cases unbiased transmission (with $c_{tresh} \leq 5$) generates samples with diversity profiles that are slightly uncharacteristic and match profiles generated under frequency-dependent transmission well.

While this could be taken as evidence for better inferential properties of the ML approach (unbiased samples are recognised as unbiased—independent of the age constraint—with a higher probability compared to the Ewens-Watterson test) we also tested how well non-neutral processes are detected. Fig 7 highlights that for low values of $p_{death}$ and $c_{thresh}$ the ML approach cannot detect a strong signature of biased transmission: for almost all $b$-values samples are considered to be neutral with a relatively high probability. However, increasing either $c_{thresh}$ or $p_{death}$ increases the reliability of the test: we see a U-shaped rejection profile which gets more pronounced for higher values of $c_{thresh}$ and/or $p_{death}$.

Fig 7 highlights clearly that there are limits to inference from population-level data at a single point in time, regardless of the approach used. Comparing this figure with Fig 2 where no age structure was assumed we see that an (unknown) age structure in combination with low $c_{thresh}$-values can mask the signature of biased transmission greatly.

Interestingly, knowledge about the age structure, i.e. the $p_{death}$ value of the population considered, does not impact the accuracy of the inference greatly (see Fig I in S1 Text) but knowledge about the $c_{thresh}$ does increase the accuracy substantially. This aligns with the finding, described above, that it is the interaction between the cultural transmission process and the age structure that mainly contributes to the potential differences between age-structured and non-age-structured population or between age-structured populations using different transmission processes.

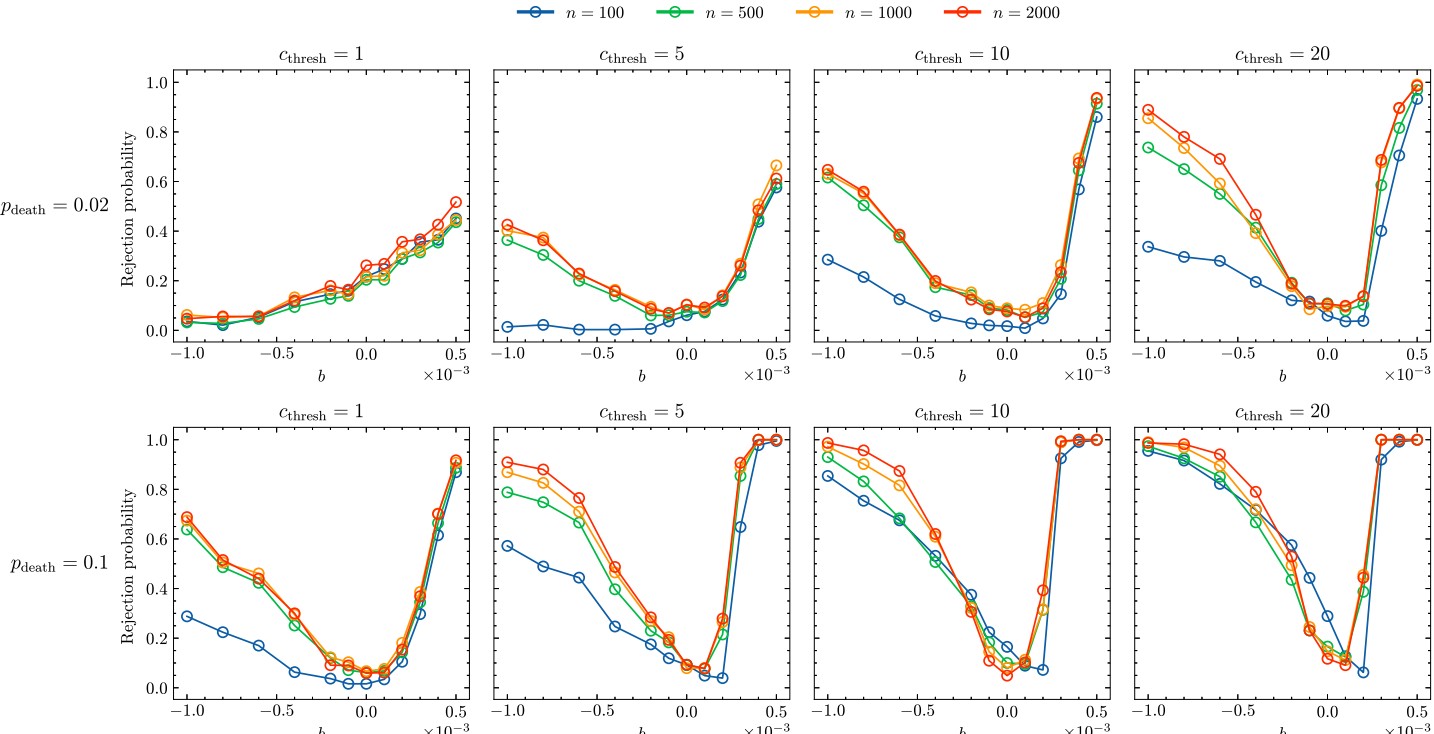

**Fig 7. Fraction of samples classified as non-neutral (by the ML approach) taken from an age-structured population with $N = 10^5$, $\mu = 5 \cdot 10^{-4}$ and various $p_{\text{death}}$ ($p_{\text{death}} = 0.02$, $p_{\text{death}} = 0.1$) and $c_{\text{thresh}}$ ($c_{\text{thresh}} = 1$, $c_{\text{thresh}} = 5$, $c_{\text{thresh}} = 10$, $c_{\text{thresh}} = 20$) values.** The different coloured lines represent different sample sizes: $n = 100$ (blue lines), $n = 500$ (green lines), $n = 1000$ (orange lines), $n = 2000$ (red lines).

## 3 Discussion

In this paper we explored the role of age structure in shaping the cultural composition of a population when cultural transmission occurs in an unbiased way. We developed an age-structured cultural transmission model in which individuals have an average life time of $1/p_{\text{death}}$ and new individuals acquire their cultural variant by copying a role model. This role model is randomly chosen from a specified copy pool determined by the parameter $c_{\text{thresh}}$ which describes the maximum age of the individuals it contains. In this way we allowed for an interplay between the age structure and the cultural transmission process leading to a structural constraint on the copy pool. We used this simulation model to generate population- and sample-level statistics for different age structures, as controlled by the value of $p_{\text{death}}$, as well as different sizes of the copy pool, as controlled by the values of $c_{\text{thresh}}$, to understand the impact of age structure on the cultural evolutionary process.

We found that the existence of an age structure *per se* has little impact on the cultural composition of the population. The 'ALL' scenario—where naive individuals can adopt cultural variants from the whole population (or from a sufficiently large fraction)—differs from the non age-structured Wright-Fisher approximation only in the shape of the variant abundance distribution and the number of variant types present in the populations. But these differences on the population level do not percolate to the observed sample level, i.e. the sample-level statistics generated by the 'ALL' scenario and the corresponding Wright-Fisher approximations had almost identical distributions. However, the interplay between cultural transmission and age structure can generate drastic differences. The '1' scenario—where naive individuals can adopt cultural variants from individuals of age 1 (or more generally from individuals of small

ages) only—produces population- and sample-level statistics that differ substantially from their Wright-Fisher approximation.

We note, too, that we aimed to understand the mechanics of the interactions between age structure and cultural evolution, and so examined only a small number of simple and extreme cases. For example, where individuals could learn from the whole population or from a small fraction of the population. In the latter case we chose to focus on transmission of the most recently copied cultural variants (the '1' scenario) which may be consistent with, for instance, the widely-studied example of baby name transmission [43] or transmission of archaeological traits over long periods of time [44]. However, it is clear that the interaction between population age structure and human cultural transmission is considerably more complex, and is itself age-dependent [15, 45]. Such empirically driven 'life histories of learning' could be fruitfully modelled in this framework, along with many other plausible 'constraints' on pools of potential role models.

However, the simple cases we present can help elucidate what, precisely, is causing the different results in situations where naive individuals can adopt cultural variants from the whole population (or a large fraction of the population) and situations where they can adopt cultural variants from a small fraction of the population only. It is an aggregation phenomenon: when the copy pool is restricted to individuals of low ages then only a small fraction of the population is contributing to the cultural dynamic. Anyone older than $c_{thresh}$ cannot be copied anymore but still contributes to the cultural composition of the population until its death. This phenomenon is known as 'time-averaging' in the archaeological literature (e.g. [44, 46–48]); cultural remains deposited at different times come to be preserved together (e.g. [49, 50]). Premo [44] demonstrated that the Ewens-Watterson test does not reliably identify unbiased cultural transmission in moderately to severely time-averaged simulated archaeological assemblages of cultural variants. This kind of assemblage can be generated by our age-structured cultural transmission model if the death rate is set to $p_{death} = 0$ and the simulation, after the burn-in period, is run for a small number of time steps. Hence the situation considered by Premo is consistent with our '1' scenario. This means our findings suggest that, in agreement with [44], time-averaging in the sense of our '1' scenario greatly impacts our ability to correctly detect biased transmission when using standard population genetics tests such as Ewens-Watterson test. However, we have also shown that time-averaging may not be problematic in situations where the cultural variants generated at very different times can still be copied. Naturally, the time horizon over which 'real' archaeological assemblages are constructed may differ substantially from the age-structured populations considered here [51] but nevertheless the question of whether older artefacts are available as a target of copying may determine whether time averaging severely impacts inferential accuracy. Whether it can be established that the excavated assemblage has been generated by a dynamic more closely resembling the '1' scenario or the 'ALL' scenario *a priori* seems unclear.

The findings regarding the 'ALL' and '1' scenario described above, are mirrored in the performance of the Ewens-Watterson test. This test evaluates whether an observed sample of size $n$ could have been drawn from a population evolving through a Wright-Fisher dynamic. If age structure has no effect on the cultural composition of the population, we would expect that only in a fraction $\alpha$ to be rejected. This is what we obtained in the 'ALL' scenario. In the '1' scenario a large fraction of samples are rejected. This means that the Ewens-Watterson test is able to reliably detect deviations from the Wright-Fisher dynamic. However, in the case of the '1' scenario those deviation are not caused by biased transmission but by the interplay between age structure and the unbiased transmission process, which induces a strong age constraint on the pool of potential role models. Consequently, test results need to be cautiously interpreted: rejections of the Ewens-Watterson test only point to a deviation from the Wright-Fisher

dynamic which—given the simplicity of the model—can be caused by a number factors, including but not exclusively by biased transmission.

To begin to tackle this problem we explored the suitability of a generative inference approach to simultaneously infer the interplay between an unknown age structure and cultural transmission, and the specific nature of the cultural transmission process. We used the age-structured cultural transmission model to generate theoretical expectations for populations with more complex demographic properties than incorporated in the Wright-Fisher model under unbiased and biased transmission processes. In particular, we simulated populations with different age structures and sizes of the copy pool to obtain their diversity profiles, i.e. Hill numbers up to order 3, under unbiased and biased (i.e. frequency-dependent) cultural transmission. We then applied machine learning techniques to classify a given sample of size $n$ as consistent or inconsistent with the hypothesis of unbiased transmission. Random forest classifiers 'learnt' the mapping between the shape of the diversity profiles (regardless of age structure and size of copy pool) and unbiased and biased transmission, respectively. This mapping function, generated from the training data, can then be applied to observed data providing the classification of the sample as biased or unbiased.

We applied this machine learning approach to the same data sets we analysed with the Ewens-Watterson test. Through the explicit modelling of age structure together with potential age constraints of the copy pool we observed an overall decrease of the probability of rejecting unbiased transmission. In sum, the machine learning approach allows us to pull apart changes in the cultural composition caused by a structural constraint on the copy pool from changes caused by biased transmission. Especially for higher $p_{death}$- and $c_{tresh}$-values we achieve reliable inference results which are less strongly impacted by the size of the sample than the Ewens-Watterson test.

Nevertheless, we also observed clear limits to inference from population-level data at a single point in time: an unknown age structure in combination with low $c_{thresh}$ values can greatly mask the signature of biased transmission. In these situations, the cultural accumulation effect is strongest. In any given time step the cultural change induced by the adoption decisions of naive individuals does not impact the cultural composition of the population greatly. This naturally invites caution when interpreting inference results based on cultural data at a single point in time. Although we can add demographic complexity to the generative model, the data simply may not be detailed enough to inform on all the different processes that generated it.

In summary, we have shown that the cultural composition of age-structured human populations evolving through unbiased cultural transmission may deviate from well-known neutral expectations in some important aspects such as the shape of the variant abundance distribution. Consequently, in many cases, modelling approaches to cultural change as well as inferential analyses must consider demographic properties of the population. Finally, our results indicate that inferential analyses may require more than cross-sectional data on the cultural composition of the population to allow for accurate inferences about transmission processes to be made. For example, when available we should move from data from a single point in time to more detailed, longitudinal data on the cultural composition of a population. Such time series data allows inference techniques to exploit more complex feature interactions, and potentially more accurately infer the cultural dynamics underlying them.

## Supporting information

**S1 Text. This supplementary material contains additional analyses of the '1' and 'ALL' scenario and further explanations of methodologies used in the paper.** In particular, section 1 describes the properties of population-level and sample-level statistics, used in this paper, for

the '1' and 'ALL' scenario. Section 2 describes an algorithm for sampling from Ewens sampling formula. Section 3 provides a comparison between the Ewens-Watterson test and the machine learning approach. Section 4 contains details on how we calculate the effective population size. Finally, section 5 provides more information about the machine learning approach.
(PDF)

## Author Contributions

**Conceptualization:** Anne Kandler.

**Formal analysis:** Anne Kandler, Laurel Fogarty, Folgert Karsdorp.

**Investigation:** Anne Kandler, Laurel Fogarty, Folgert Karsdorp.

**Methodology:** Anne Kandler, Laurel Fogarty, Folgert Karsdorp.

**Software:** Folgert Karsdorp.

**Writing – original draft:** Anne Kandler, Laurel Fogarty, Folgert Karsdorp.

**Writing – review & editing:** Anne Kandler, Laurel Fogarty, Folgert Karsdorp.

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
