## [Decision Letter · Decision Letter 0]

1 Mar 2023

Dear Dr Kandler,

Thank you very much for submitting your manuscript "The interplay between age structure and cultural transmission" for consideration at PLOS Computational Biology.

As with all papers reviewed by the journal, your manuscript was reviewed by members of the editorial board and by several independent reviewers. In light of the reviews (below this email), we would like to invite the resubmission of a significantly-revised version that takes into account the reviewers' comments.

Below you find the detailed feedback of three expert reviewers. They all agree that the manuscript makes a very valuable contribution to the literature.

However, they also point out a few instances where the manuscript can be further improved.

For example, both reviewers 1 and 2 suggest that the important concepts of neutrality and bias are used somewhat ambiguously.

In addition, reviewer 3 suggests that some model assumptions should be discussed and motivated in more detail.

I believe all the review reports are extremely constructive and I am sure the authors will find the feedback very valuable.

Please address all the reviewers' comments in the revised version.

We cannot make any decision about publication until we have seen the revised manuscript and your response to the reviewers' comments. Your revised manuscript is also likely to be sent to reviewers for further evaluation.

Sincerely,

Christian Hilbe

Academic Editor

PLOS Computational Biology

Natalia Komarova

Section Editor

PLOS Computational Biology

Below you find the detailed feedback of three expert reviewers. They all agree that the manuscript makes a very valuable contribution to the literature.

However, they also point out a few instances where the manuscript can be further improved.

For example, both reviewers 1 and 2 suggest that the important concepts of neutrality and bias are used somewhat ambiguously.

In addition, reviewer 3 suggests that some model assumptions should be discussed and motivated in more detail.

I believe all the review reports are extremely constructive and I am sure the authors will find the feedback very valuable.

Please address all the reviewers' comments in the revised version.

Reviewer's Responses to Questions

**Comments to the Authors:**

Reviewer #1: In “The interplay between age structure and cultural transmission,” the authors develop an age-structured model of cultural transmission (which may be unbiased or frequency-dependent biased); advance a machine learning (ML) test of unbiased vs. biased cultural transmission; and compare this ML approach with the Ewens-Watterson test on simulated data from age-structured and non-age-structured models. Among other findings, they show that in age-structured populations where newborn individuals randomly copy the cultural variants of sufficiently young individuals, the ML approach outperforms the Ewens-Watterson test in classifying such transmission as ‘unbiased.’ Overall, while I have several minor suggestions and one major question, I believe this paper -- particularly the ML approach developed -- is highly valuable to the field of cultural evolution and will have a significant impact. Further, this paper was a pleasure to read, technical details were explained well, and results were presented clearly.

My major question pertains to the definition of neutrality. Unbiased transmission is used synonymously with neutrality (e.g., line 54), while biased cultural transmission = non-neutrality = ‘selection’ in the paper. However, in many places the authors state or imply that a bias which they call “recency bias” should be recognized as neutral. For instance, line 396: “neutral samples are recognised as neutral — independent of the assumed age structure — with a higher probability compared to the Ewens-Watterson test…” and these ‘neutral samples’ are taken from populations with the recency bias (c_thresh = 1,5,...) (but are neutral in the sense that there is no frequency-dependent bias: b = 0). Also see lines 362-363.

Fixing this issue likely is not as simple as removing the word “bias” from “recency bias” and calling it something like age-structured-unbiased-transmission. More fundamentally it needs to be explained why some biases, like frequency-dependent bias, are considered non-neutral and should be classified as such by Ewens-Watterson, but recency bias should be “recognized as neutral.” My first thought was that recency bias involves random copying from a subset of the population while frequency-dependent bias involves non-random copying. In that case, would demonstrator bias (where a randomly selected subset of individuals in the population are copied randomly by naive individuals) also count as neutral? But what if this randomly selected copy pool included just 1 individual (one-to-many transmission)? Another idea is to imagine that in recency bias, individuals have zero preference / bias / tendency to copy young individuals over older individuals, but are only able to observe young individuals’ traits because of some structural features in the population (implied on lines 441-444). However, I don’t immediately see what these features would be -- or how the conceptual re-framing of a cultural transmission bias to a structural feature without a mathematical distinction between them means that Ewens-Watterson gives “erroneous inference results as cultural change caused by the structural features of the populaton is (wrongly) attributed to changes in the underlying transmission process” (lines 442-444). To sum up, I wonder whether it may be more appropriate to say that the Ewens-Watterson test correctly fails to classify samples with a strong recency bias as neutral. And rather than ‘out-performing’ Ewens-Watterson, the ML approach can simply tackle different questions -- such as distinguishing b=0 from b=/=0 in populations with recency bias.

Some other comments / questions:

-I wondered why p_death did not depend on age -- intuitively, in a given generation, very old individuals might be more likely to die than younger individuals.

-Line 111 says that with probability mu, a new and never-before-seen cultural variant is introduced, but line 122 says that each simulation reaches equilibrium. How can an equilibrium (no change in frequency of variants) be reached if new variants are constantly being introduced (going from 0 to + frequency)?

-Also, in a finite population, polymorphic equilibria do not occur; there would be fixation of only one variant at equilibrium. This doesn’t seem to be happening here (e.g., line 259) suggesting that ‘equilibrium’ may not be the right term.

-In Figure 1B and 1D, adding x- and y- axis labels would be helpful

-In Figure 1C, the top-right circle with D <= 1 had a formatting issue

-Table 1 shows N_e but what was the original N used for this calculation?

-In the captions of Figure 4 and Figure 5 it could be clarified that these are the results from the Ewens-Watterson test and that b = 0

-In the captions of Figure 6 and Figure 7 it could be mentioned that these results are from the ML model

In closing, overall, this paper was enjoyable to read and presented an exciting new approach for interpreting data from age-structured populations. I look forward to the authors’ reply and hope these comments are useful to them.

Reviewer #2: Review of PCOMPBIOL-D-22-01705

The submission is a theoretical/computational study in cultural evolution. The focus is on our ability to infer signatures of selection among cultural variants from cross-sectional (not longitudinal) samples of frequencies of these cultural variants. In particular, the authors look at what happens when cultural variants are from an age-structured population (i.e. generations are overlapping and individuals may have different ages) and we use tools that are not specifically designed for age-structured populations to make statistical inferences (i.e. to reach conclusion about selection VS no selection) from samples taken from this population.

The tools under considerations are two. One is the Ewens-Watterson test (which has sometimes been applied in the context of unveiling the presence of cultural selection from archeological data), the other is a machine learning technique developed by the authors. I will comment neither about the construction nor about the (comparative) performances of the machine-learning tool, as this would be far from my expertise.

To approach the problem, the authors set up a cultural-demographic model in discrete time. They consider a population where survival of individuals from one time point to the next (and thus from one age to the next) is a probability p_{death} that is independent of age. Constant population size is ensured by introducing at each time point new recruits in number equal to the individuals who just died. Transmission of cultural variants can depend on age. The authors deem relevant the case in which transmission can only depend on the age of the cultural parent (i.e. the individual from which culture is acquired). A parameter (c_{threshold}) sets the age below which individuals can transmit their cultural variant. Cultural transmission never depends on the age of the cultural offspring (the individual acquiring culture), who always are the new recruits (age 0) to the population. Barring a mutation event (happening with probability \\mu per individual per time step) that confers a new recruit a new cultural variant previously unseen in the population (akin to an infinite allele model), non-mutated recruits adopt for their lifetime the cultural variant copied from some individual aged less than c_{threshold} who already is in the population.

The authors distinguish between unbiased transmission and biased transmission. What the authors call unbiased transmission is defined by their Eq. 1. Unbiased transmission occurs when, in the absence of a mutation, the probability of picking up cultural variant k is proportional to the current frequency of k in the population from which variants can be chosen from. Biased transmission occurs when this probability deviates from current frequency of the variant (Eq. 2). The authors appear to equate unbiased transmission with neutrality (absence of selection) and biased transmission with selection.

One of the main results of the authors is that, under unbiased cultural transmission (i.e. cultural neutrality), the performances of the Ewens-Watterson test on cross-sectional samples from numerical simulations of their model are good when new recruits can acquire their cultural variant from any individual in the population. However, while keeping cultural transmission unbiased and thus the population neutral, performances of the test get poorer the lower is the age (c_{threshold}) beyond which individuals cannot transmit their cultural traits anymore. Low mortality can further exacerbate this situation. See in particular their Figs. 4-5. Therein, the probability of rejecting the hypothesis of neutrality despite transmission remaining unbiased increases with both diminishing c_{threshold} and diminishing p_{death}. The reason is that individuals of age c_{threshold} or more are culturally post-reproductive or inert. Hence, individuals actually involved in the dynamics of cultural transmission (the effective size) only are a subset of the total population size. And this subset gets smaller the lower the death probability because this allows culturally inert individual to persist longer in the population. Samples from which inferences about selection should be made, however, are taken from the total population, and not only from its effective fraction, thereby including inert individuals in proportion to their representation in the total population. The statistical test applied to such samples is not equipped to deal with discrepancies between effective and actual population size, as the test is built on the basis of the Wright-Fisher model where effective size and actual size coincide. (The very notion of effective size in population genetics roughly is the size a Wright-Fisher population would have to show the same neutral dynamics as a population which dynamics deviate from Wright-Fisher assumptions. For this reason, the authors go on calculating the effective size for their model and make some comparisons based on that as well.)

From this result, the authors draw one of their main conclusions: if researchers apply this test to cultural datasets, which are samples of a larger evolving culture, they may incorrectly infer the presence of selection in this culture whereas actually there is none. What these researchers could mistake for signs of selections would be mere artefacts due to unbiased cultural transmission tied to a restricted age range (especially when comprising young ages) of cultural parenting.

Overall, I am sympathetic with what seems to be a key aim of the paper, alerting about the dangers of uncritically applying theoretical methods derived for unstructured populations to structured populations. The specific case of the authors is age structure and cultural evolution, but the same moral has already been shown to extend to other forms of structure and biological evolution. In fact, I would urge the authors to refer in their work to those papers that have shown how population structure can lead the values of neutral quantities like the neutral substitution rate to deviate from their values as expected from models assuming unstructured populations. As for culture, the authors focus on age structure and link cultural transmission to age. But I can also imagine that other forms of population structure (spatial, where individuals are separated in space; demographic, where individuals can be of different stages; cultural, where individuals have different levels of cumulated culture or cultural transmission abilities) can produce results similar to those obtained here. At bottom, population structure of any sort tends to make effective size different from census size and, therefore, predictions based on Wright-Fisher assumptions may fail to extrapolate to structured populations.

I have one main concern of conceptual nature that shows where the submission may fail to drive its point home. The authors equate unbiased cultural transmission with selective neutrality and biased cultural transmission (any deviation from unbiased) with selection. In particular, the authors consider forms of frequency-dependent selection. In their definition of unbiased cultural transmission (Eq. 1), they set the transmission probability (without mutation) equal to the frequency of the cultural variant in the culturally active population only (i.e., those individuals of age less than c_{threshold}), and not in the total population. I believe that this definition is narrow and it actually includes bias in the form of age bias, because young individuals are culturally fitter than old ones. So when the pop-gen test for neutrality is applied to a sample from the total population and the test tends to reject the neutrality hypothesis especially when cultural fitness is highly concentrated into young ages, one could conclude that the test results are in principle correct: the test tends to detect a bias in cultural transmission that is there, although this bias is not at the level of cultural variants (they are not of different fitnesses) but at the level of age (different ages may have different fitnesses). In a sense, the test does its job if the job is to detect bias. Note also that cultural dynamics in the model are frequency dependent (in the frequency of ages, of course) if we assume what the authors define as unbiased transmission, as cultural dynamics depend on the changing age distribution of the population. So when the authors consider selection to occur in its frequency-dependent form, one gets easily confused. Hence, the reader is left with the impression that important concepts (bias, neutrality, selection and frequency dependence) get mixed throughout the submission. The authors should clarify well and keep separate all the different aspects of their work.

Minor remarks:

Line 31, no doubt the work of Norton (1931) was definitely seminal/foundational to the field. However, nowadays we can think of this work as basically being absorbed by Charlesworth book on age-structured populations you refer to later

Line 32, in Leslie (1945) the word ‘stage’ doesn’t appear once. In fact, the famous Leslie matrix model is classified only by age and not by stage. If the authors believe it is really relevant here to mention how stage and age descriptions can be demographically intertwined, a more appropriate reference perhaps is Caswell et al (2018) Age x Stage-classified demographic analysis, EcoMono

Line 39, to explain

Line 42, Hewlett and Cavalli-Sforza [14] showed

Lines 104, the way I understand the model from the previous lines is that there is age-independent survival and no maximum age. Here, however, it would seem that there is a maximum age a_{max}. I suppose that a_{max} actually is the age of the most longevous individual currently in the population, so that a_{max} is actually a function of time t and not some exogenously imposed parameter.

Lines 107, as far as I can tell, the cultural variants are assumed never to alter vital rates of individuals. The formulation in this sentence gives the wrong suggestion that there may be a moment in which this assumption is relaxed.

Reviewer #3: In general, this manuscript is nearly publishable as-is, but perhaps some minor revisions would be useful.

The authors address an important issue in studying social learning and cultural transmission – social learning is not always random, but structured. We learn skills from those who have the skills, younger individuals initially learn from older ones, whether that is from parents or older siblings. And it is true that quantitative modeling in the cultural transmission literature has mostly ignored the effects of age structure, so the authors perform a valuable service by proposing one type of age-structured model and examining our ability to identify that model in sample data as compared to the standard null model of unbiased transmission.

One question that seems unanswered in the paper is the problem that led to this research: what motivated them to conduct this research? Most realistic genetics models for complex species are age-structured, so it's reasonable to wonder how age-structuring might work. But is there a particular problem in existing work that this new model is designed to resolve? Are there particular cases for which the age-structuring would be confounding (are they rare cases? Common cases? A certain class of problems?) The article might be strengthened by starting with a model in which individuals that are newly born copy from parents, and then at some threshold age, they begin to copy from the general population perhaps – an abstract version of child development. So one question might be: does child development have an effect on diversity and richness? Or perhaps there could be a discussion of the scenarios where the traditional assumptions about the nature of sampling during social learning likely don’t hold true and thus could present challenges.

With that said, one area may benefit from elaboration, to put the modeling exercise into context with the literature on social learning. In formulating their age-structured model, the authors make a modeling choice to represent “age structuring” simply as a restriction on the maximum age of the effective pool of models. In other words, their model is tunable to the extent that it can represent populations where individuals only copy from some slice of the youngest individuals – up to the limiting model (called “1” in the manuscript) where individuals only copy from the most recently “born”. But we know that social learning, especially in humans and non-human primates, starts out being age-structured in the opposite way: infants spend the most time initially learning from older individuals, principally the mother. Gradually, the pool of cultural models expands as children become mobile and join group activities, with the pool of models expanding to include age-peers and older children, along with some contact with a variety of adults. Later, “oblique” transmission between younger and older individuals occurs in the context of mentorship, apprenticeship, and formal “teaching”.

There is, of course, considerable variation in these patterns, but they are well documented in the observational literature on primate social learning and humans' social and developmental psychology. The authors do not discuss this in any detail, and the article would benefit from the additional context on what patterns of age structure do empirically occur, and which matter most. Unfortunately, such a review would tend to reveal that the modeling choice made by the authors (to uniformly restrict copying to the N youngest “slices” of the population effectively) is a strange one.

The modeling choice may have been made for convenience, but later discussion of “time averaging” in the discussion also suggests that the choice of allowing copying of the youngest cohorts may have been done to make a link between “time averaging” effects and the inability to distinguish different models – a subject previously studied by Premo and Madsen.

We are not suggesting that the authors redo their modeling, but at least give us a reason why this form of age-structuring is relevant to applications (if indeed it is), rather than age structures that more closely replicate what we see observationally in living populations.

One point that might be explored: If one is always sampling from the youngest cohort, one might expect increased diversity because that's where all the "fresh" mutations are (by the construction of the model). As we know in unbiased transmission/WF sampling, once a new variant is created, it inevitably starts to decay out of the population unless it's very lucky. But this effect would be partially balanced out by some number of lingering "older" mutations that are still in the population. Since WF sampling from a fixed-size population is sort of like calculating the balance between water coming out of a hose into a pool and the pool draining over the rim, you should be able to calculate the degree to which model "1" affects the mean richness.

A final note is that the authors claim that the machine learning technique for studying equifinality among unbiased and biased models – random forest classifiers – is “novel” in the literature. This is only partially true. In previous work, given at conferences and in his 2020 dissertation, Madsen used random forest classifiers in exactly this fashion and proposed it as a general technique for generatively testing whether models are empirically distinguishable (Chapter 3). That said, the work in Chapter 3 does not appear to have been published in journal form at this point, so it is understandable that the authors may not be aware of it.

In summary, this is a valuable contribution to the ongoing effort to model cultural transmission and social learning in a rigorous, mathematical manner, and to develop statistical tools to fit models. The manuscript could be improved slightly to include more context on the reason for modeling choices and how they relate to known patterns of social learning, but otherwise should be published.

**Have the authors made all data and (if applicable) computational code underlying the findings in their manuscript fully available?**

Reviewer #1: Yes

Reviewer #2: Yes

Reviewer #3: Yes

PLOS authors have the option to publish the peer review history of their article (what does this mean?). If published, this will include your full peer review and any attached files.

Reviewer #1: No

Reviewer #2: No

Reviewer #3: No
---

## [Decision Letter · Decision Letter 1]

23 Jun 2023

Dear Dr Kandler,

We are pleased to inform you that your manuscript 'The interplay between age structure and cultural transmission' has been provisionally accepted for publication in PLOS Computational Biology.

Best regards,

Christian Hilbe

Academic Editor

PLOS Computational Biology

Natalia Komarova

Section Editor

PLOS Computational Biology

In the meantime, we have got feedback from two of the original three reviewers (Reviewers #2 and #3). They both confirm that the authors have improved the manuscript substantially, and that it should be accepted. I fully agree. I would like to thank the authors for their careful revision. Please take the remaining comments of Reviewer #2 into account when uploading the final document.

Reviewer's Responses to Questions

**Comments to the Authors:**

Reviewer #2: Review of PCOMPBIOL-D-22-01705R1

I would like to thank the authors for their efforts to revise the manuscript according to the comments.

I have no further comment on the current version of the manuscript except the minor points below:

Minor points:

1. Author summary in first single person in contrast to abstract and main text

2. Line 83, “populations have no significant structure”

3. Line 116, to explore

4. Throughout, inconsistency between ‘All’ and ‘ALL’

5. Ref 27 starts with comma?

Reviewer #3: The authors have done a great job of addressing all of the suggestions and comments of the reviewers. I have no further suggestions and recommend that the paper be accepted for publication.

**Have the authors made all data and (if applicable) computational code underlying the findings in their manuscript fully available?**

Reviewer #2: Yes

Reviewer #3: Yes

PLOS authors have the option to publish the peer review history of their article (what does this mean?). If published, this will include your full peer review and any attached files.

Reviewer #2: No

Reviewer #3: **Yes: **Carl Lipo

---

## [Editor Report · Acceptance letter]

8 Jul 2023

PCOMPBIOL-D-22-01705R1 

The interplay between age structure and cultural transmission

Dear Dr Kandler,

I am pleased to inform you that your manuscript has been formally accepted for publication in PLOS Computational Biology. Your manuscript is now with our production department and you will be notified of the publication date in due course.

With kind regards,

Zsofia Freund
